# Using artificial neural networks to reveal the human confidence computation

Medha Shekhar[1,2]*, Herrick Fung[1], Krish Saxena[1], Farshad Rafiei[1], Dobromir Rahnev[1]

1 School of Psychology, Georgia Institute of Technology, Atlanta, Georgia, United States of America,
2 Center for Research in Cognition & Neurosciences, ULB Neuroscience Institute, Université libre de Bruxelles, Brussels, Belgium

* medha.shekhar@ulb.be

## Abstract

Humans can evaluate the accuracy of their decisions by providing confidence judgements. Traditional cognitive models have tried to capture the mechanisms underlying this process but remain mostly limited to two-choice tasks with simple stimuli. How confidence is given for naturalistic stimuli with multiple choice alternatives is not well understood. We recently developed a convolutional neural network (CNN) model – RTNet – that exhibits several important signatures of human decision making and outperforms existing alternatives in predicting human responses. Here, we use RTNet's image computability to test seven different confidence strategies on a task involving digit discrimination with eight alternatives (N = 60). Specifically, we compared confidence strategies that consider the entire evidence distribution across choices against strategies that only consider a specific subset of the available evidence. We also compared strategies based on whether they compute confidence from posterior probabilities or from raw evidence. A model which derives confidence from the difference in raw evidence between the top-two choices (Top2Diff model) consistently provided the best quantitative and qualitative fits to the data and the best predictions of human confidence. These results support the notion that human confidence is based on a specific subset of available evidence, challenge prominent theories such as the Bayesian confidence hypothesis and the positive evidence heuristic, and establish CNNs as promising models for inferring the mechanisms underlying human confidence in naturalistic settings.

## Author summary

Human decisions are accompanied by a sense of confidence which reflects the decision accuracy. Conventionally, human confidence has been studied using two-choice tasks with simple stimuli such as Gabor patches and dots, making it unclear how confidence is given for naturalistic stimuli with multiple choice

**Data availability statement:** All data and code have been made publicly available at https://osf.io/xcv98/.

**Funding:** This study was funded by the National Institutes of Health (NIH; R01MH119189 to DR) and the Office of Naval Research (ONR; N00014-20-1-2622 to DR). The funders had no role in study design, data collection and analysis, decision to publish, or preparation of the manuscript. Salary support was provided by the NIH to MS, HF, and DR, and by the ONR to DR.

**Competing interests:** The authors have declared that no competing interests exist.

alternatives. In this study, we used a neural network model called RTNet to investigate human confidence mechanisms in an eight-choice task with handwritten digits. We instantiated seven confidence strategies that differed based on: 1) how much perceptual evidence is incorporated into confidence, and 2) whether confidence uses the raw perceptual evidence or computes probabilities from this evidence. Our results showed that human confidence was best explained by a model that derives confidence from the difference in raw evidence between the top-two choices. These results support the notion that human confidence is based on a specific subset of the raw evidence and establish a framework for inferring the mechanisms underlying human confidence in naturalistic settings.

## Introduction

As humans, we can introspect on our own decisions and provide confidence estimates that reflect the likelihood of these decisions being correct [1–3]. However, the computations behind confidence remain debated and thus revealing them is seen as one of the central medium-term goals for the field [4]. Popular cognitive modelling frameworks including signal detection theory [5–13], sequential sampling [14–17] and Bayesian inference [18–22] have all attempted to capture the mechanisms underlying confidence generation and successfully provided insights into these processes. However, there are two major limitations associated with these models. Firstly, cognitive modelling has almost exclusively been applied to tasks involving only two choices [23]. Despite notable attempts [24,22], extending these models to more complex tasks involving multiple choices has proved difficult. For instance, classic SDT is fundamentally limited to 2-choices tasks and Bayesian computations become intractable for a large number of alternatives [25]. Secondly, these models have only been applied to tasks involving simple visual stimuli such as Gabor patches and moving dots since traditional models of decision making are not image computable (i.e., they cannot process images involving complex features). Therefore, it is still unclear how confidence is generated for decisions involving naturalistic stimuli and multiple alternatives.

One promising approach for addressing this question is to use convolutional neural networks (CNNs). The advantage of CNNs is that they are image computable models of vision that are capable of human-level object recognition [26,27]. Further, they can easily be applied to decision making tasks with several alternatives. These features of CNNs can potentially be leveraged to model confidence mechanisms in tasks involving naturalistic images and multiple alternatives. However, although CNNs perform object classification tasks with high levels of accuracy, they cannot yet be considered as good models of human vision as it is unclear whether their decision-making mechanisms are similar to humans [26,27]. Particularly, their behavior has not been extensively validated on human data for tasks involving simple psychological manipulations. Therefore, standard CNNs may not yet be suitable for modelling and inferring the processes occurring in humans.

To address these challenges, we recently developed a dynamic neural network model, called RTNet, which combines the image processing capabilities of CNNs with mechanisms drawn from empirically tested cognitive models [28]. Specifically, RTNet has two modules: (1) a CNN with stochastic weights that generates noisy evidence at each time step and (2) an evidence accumulator which integrates this noisy evidence towards a threshold (**Fig 1A**). Critically, RTNet reproduces several key signatures of human perceptual decisions and also predicts human RT, choice and confidence for novel, individual images. Further, RTNet clearly outperformed other biologically inspired CNN architectures involving processes such as recurrence (BLNet; [29]), parallel processing (CNet; [30]), and resource-efficient processing (MSDNet; [31]). Based on these findings, RTNet can be considered as a promising model of human decision making that can be further applied towards examining confidence mechanisms in humans.

Previous studies have proposed a range of different strategies for computing confidence based on how much of the available perceptual evidence is incorporated into confidence. For instance, confidence can be computed using all the evidence available for the perceptual decision. A popular example of this strategy is the Softmax transformation, which is currently the standard method for computing confidence in neural networks. Softmax normalizes evidence by accounting for its spread across all choice options [32]. However, it has not been tested in humans using cognitive models. In contrast, other proposed strategies only utilize specific subsets of the available evidence. For instance, confidence may be computed as the difference in evidence between the two most likely options (Top2Diff strategy). Alternatively, another

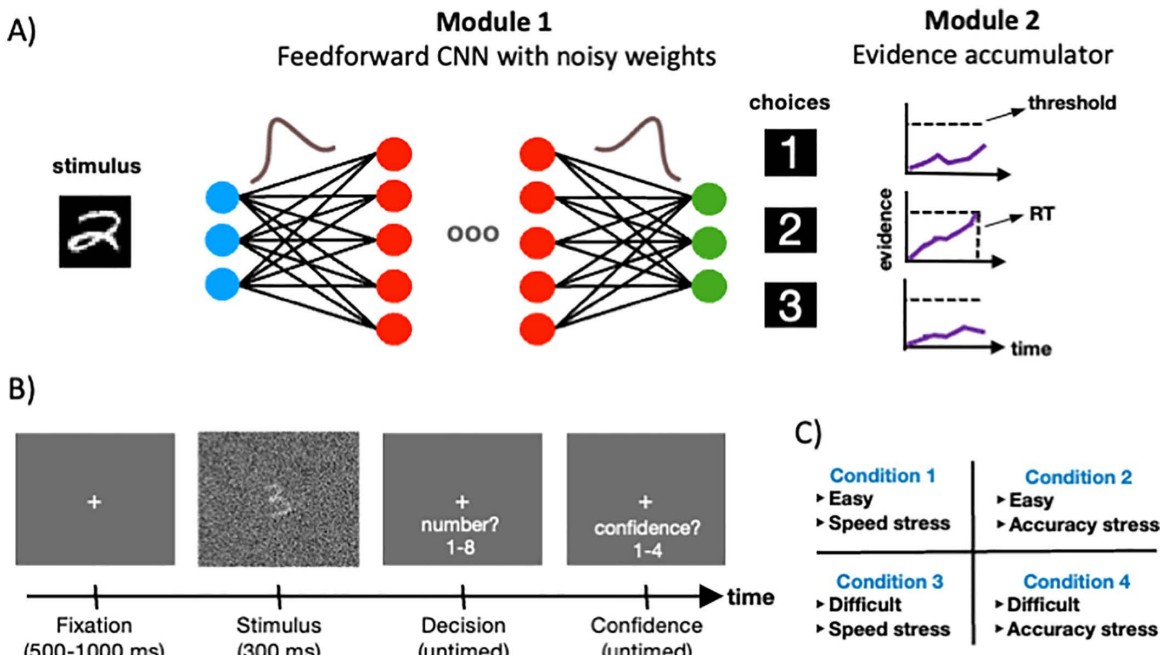

**Fig 1. RT architecture and task.** (A) RTNet architecture. RTNet consists of two modules. The first is a Bayesian neural network whose weights are stochastic such that at each processing step, a unique feedforward CNN is instantiated. As a result of the stochastic weights, the network processes the same image at each time step and generates noisy evidence through the activations of its output layer. The evidence for each choice option is then accumulated by an evidence accumulator module towards a pre-defined threshold. (B) Task. Subjects performed a digit discrimination task where they were presented with a noisy hand-written image of a digit between 1 and 8. The Subjects decided which digit was presented and then reported their confidence on a scale from 1-4. (C) The four experimental conditions. Task difficulty as well as speed-accuracy trade-off instructions were manipulated in a 2x2 factorial design. Task difficulty was adjusted by adding noise to the images, whereas speed-accuracy trade-off was manipulated by instructing subjects to focus on either accuracy or speed.

popular theory is that confidence only considers the evidence in favor of the chosen response (i.e., the "positive evidence"), neglecting all evidence against the choice [25,33–36].

The three strategies mentioned above derive confidence directly from the evidence underlying the choice. However, another popular set of theories is based on the assumption that confidence relies on explicit probability computations rather than raw evidence. For instance, confidence can be computed as the Entropy of the posterior probability distribution across all available choices (Entropy strategy; [22,29]). Alternatively, Comay et al. [24] proposed that confidence is computed as the difference between the posterior probability of the most likely choice and the average of posterior probabilities of the residual choices (the ProbAvgRes model). In contrast, other strategies use only a subset of the posterior distribution. Li & Ma [22] implemented a probability-based version of the Top2Diff model where confidence is computed as the difference in posterior probabilities of the two most likely choices (the ProbTop2Diff). Finally, the Bayesian confidence hypothesis (the BCH model), one of the most popular theories in the field, proposes that confidence is the posterior probability associated with the chosen response [37,38]. Confidence in the BCH model uses the posterior probability associated with just the chosen category, and unlike the remaining probability-based models above, ignores the probability associated with all other choices.

Despite the fact that various confidence strategies have been proposed, there is currently little empirical work to distinguish between them. Perhaps the only attempts to quantitatively compare some of these strategies in the context of a multi-alternative task were made by Li & Ma [22] and Comay et al. [24]. Li & Ma [22] applied Bayesian modelling to a 3-choice task and showed that the ProbTop2Diff model outperformed both the BCH and Entropy models. However, Li & Ma's task was unique in that, unlike most of the literature, none of the alternatives were objectively correct on any given trial. Comay et al. (2023) used a similar approach to model confidence but for a three-choice task where there was indeed an objectively correct answer on every trial. Their results showed that ProbAvgRes outperformed BCH and ProbTop2Diff. However, neither of these studies examined evidence-based models, despite recent evidence that confidence computations are likely performed on the raw sensory evidence [39] Further, both of these studies used relatively simple stimuli, and both only examined three-choice tasks, leaving it unclear how confidence is computed in cases with higher number of alternatives and complex images.

Here, we compared human confidence judgments in a complex, 8-choice digit discrimination task (**Fig 1B**) with confidence ratings generated by RTNet using each of the seven confidence strategies described above: Softmax, Top2Diff, PE, Entropy, ProbAvgRes, ProbTop2Diff, and BCH. The task involved discriminating hand-written MNIST stimuli [40] and featured difficulty and speed-accuracy trade-off manipulations (**Fig 1C**). We found that the Top2Diff strategy emerged as the best model: it provided the best quantitative fits, reproduced qualitative signatures of human confidence, and predicted well the confidence pattern for correct and error choices. These results suggest that confidence in humans is computed as the difference in raw evidence between the top two options and establish CNNs as promising models for inferring mechanisms underlying human decisions in naturalistic settings.

## Results

We compared the ability of seven confidence generation strategies – Softmax, Top2Diff, PE, Entropy, ProbAvgRes, ProbTop2Diff, and BCH – to fit human confidence in an 8-choice digit discrimination task. Each strategy was implemented within our recently developed neural network model, RTNet [28]. The RTNet architecture consists of two modules – (1) a Bayesian neural network (BNN) with stochastic weights and (2) an evidence accumulator. The stochastic weights of the BNN generate a unique feedforward network at each time step and lead to variable activations in the network's final layer. The evidence accumulator receives these noisy activations from the BNN and integrates the evidence towards a pre-defined threshold. Evidence is accumulated independently for each choice option and the model's decision corresponds to the option for which the evidence first crosses the threshold.

In our previous work, we had trained RTNet on the MNIST dataset and then fit the noise and boundary parameters (see Methods) to match human performance for difficulty and speed-accuracy trade-off manipulations [28]. Here, we

 

fit the same pre-trained RTNet model to the human confidence ratings using the activations at the final time-step when the model generated the decision. Specifically, for each confidence strategy, we fit a set of three confidence criteria that transform the continuous confidence decision variable for each confidence strategy into 4-point confidence ratings. We fit each model to the data of each human subject separately. We then assessed the quality of model fits and the predictions of each strategy for the confidence associated with previously unseen individual images.

## Quantitative model comparisons

We first compared how well the seven confidence strategies fit the observed data using AIC scores. Lower AIC scores indicate better fits to the data. AIC comparisons revealed that the Top2Diff strategy significantly outperformed all other strategies by generating the lowest AIC values (**Fig 2A**). Specifically, Top2Diff outperformed the ProbAvgRes strategy by an average of 11.64 points (95% CI = [4.26, 18.19]), the ProbTop2Diff strategy by 15.66 points (95% CI = [7.57, 24.24]), the Entropy strategy by 22.43 points (95% CI = [11.69, 35.95]), the PE strategy by 58.68 points (95% CI = [46.74, 76.74]), the Softmax strategy by 69.79 points (95% CI = [51.28, 103.34]), and the BCH strategy by 80.59 (95% CI = [60.61, 108.77]) points. These differences correspond to the Top2Diff strategy being, on average, 337 times more likely than ProbAvgRes, 2521 times more likely than ProbTop2Diff, $7.4 \times 10^4$ times more likely than Entropy, $5.3 \times 10^{12}$ times more likely than PE, $1.4 \times 10^{15}$ times more likely than Softmax, and $3.2 \times 10^{17}$ times more likely than BCH. In other words, the Top2Diff strategy provided a much better quantitative fit to the human data compared to any of the other strategies.

## Qualitative model comparisons

To gain further insight into the performance of the seven confidence strategies, we compared how well each strategy can fit different qualitative patterns of confidence. Specifically, we examined the effects of the two behavioral manipulations

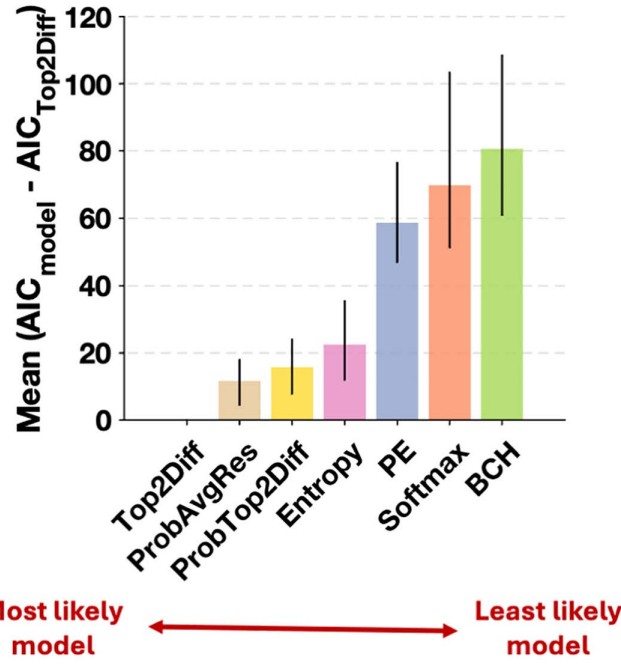

**Fig 2. Comparing the ability of confidence strategies to fit human confidence ratings.** We examined the quantitative fits of the seven confidence strategies (PE, BCH, Top2Diff, ProbTop2Diff, ProbAvgRes, Entropy and Softmax). The Top2Diff strategy outperformed all other confidence strategies by an average of at least 11 AIC points. Error bars depict 95% confidence intervals for the difference in AIC scores between Top2Diff and each of the remaining strategies.

PLOS Computational Biology

(task difficulty and speed-accuracy trade-off) on confidence. In addition, we examined the folded-X pattern which is popularly regarded as a critical signature of human confidence [37,38].

First, we evaluated the effect of task difficulty manipulation on confidence. For humans, we found that confidence decreased from 3.50 in the easy to 3.14 in the difficult condition ($t(59) = 11.768$, $p < 0.001$; **Fig 3A**). All strategies could replicate this general pattern (Top2Diff: easy = 3.49, difficult = 3.26; ProbAvgRes: easy = 3.50, difficult = 3.22; ProbTop2Diff: easy = 3.50, difficult = 3.24; Entropy: easy = 3.47, difficult = 3.28; PE: easy = 3.41, difficult = 3.31; Softmax: easy = 3.50, difficult = 3.28; BCH: easy = 3.54, difficult = 3.27). Numerically, the ProbTop2Diff strategy produced the smallest error magnitude as measured by the average sum of squared errors across subjects (Top2Diff = 0.041; ProbAvgRes = 0.040;

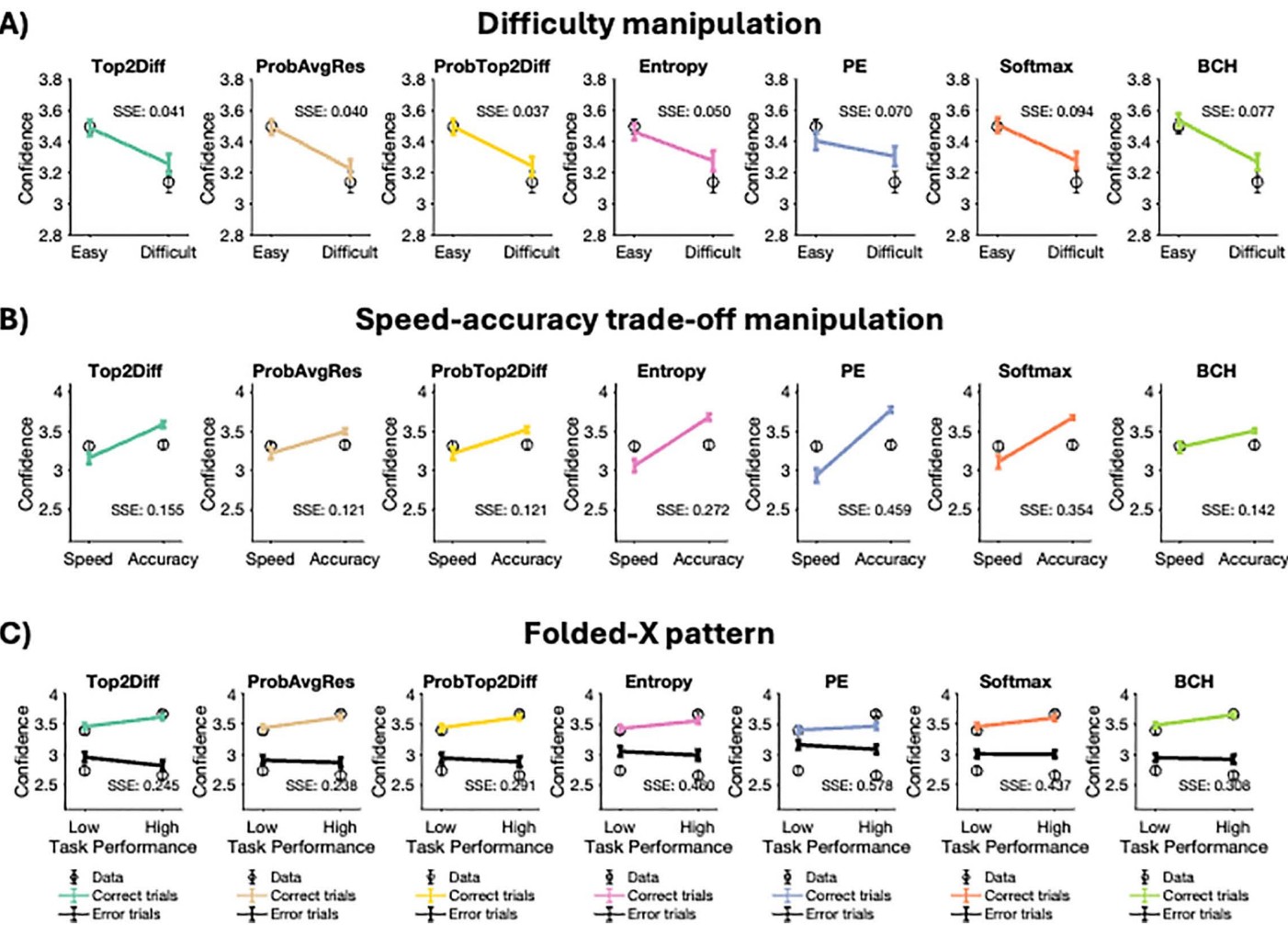

**Fig 3. Comparing the ability of confidence strategies to fit patterns of human confidence.** We examined the qualitative fits of the seven confidence strategies. (A) Confidence decreases with task difficulty. All confidence strategies were able to reproduce this qualitative pattern, but the Top2Diff, ProbAvgRes and ProbTop2Diff models provided the closest fits to the data. (B) There was no significant change in confidence between the speed and accuracy focus conditions. All models incorrectly predicted that confidence should be higher in the accuracy- compared to the speed-focus condition, but ProbAvgRes and ProbTop2Diff models provided the closest fits to the data. (C) Confidence increased with task performance for correct trials but decreased with task performance for error trials giving rise to a folded-X pattern. All strategies except Entropy can reproduce this qualitative pattern, but the Top2Diff, ProbAvgRes and ProbTop2Diff models provided the closest fits to the data. Error bars depict SEM. SSE, sum of squared errors (smaller values indicate better fits).

ProbTop2Diff = 0.037; Entropy = 0.050; PE = 0.070; Softmax = 0.094; BCH = 0.077). However, these errors were not significantly different from the errors generated by the Top2Diff and ProbAvgRes models (Top2Diff: t(59) = 1.094, p = 0.28; ProbAvgRes: t(59) = 0.617, p = 0.54). The remaining strategies – Entropy, PE, Softmax and BCH produced significantly worse fits compared to ProbTop2Diff (Entropy: t(59) = 2.278, p = 0.026; PE: t(59) = 4.974, p < 0.001; Softmax: t(59) = 3.342, p = 0.001; BCH: t(59) = 3.415, p = 0.019).

Second, we evaluated the effect of the speed-accuracy trade-off manipulation on confidence. In humans, there was no significant change in confidence from the speed-focus to the accuracy-focus condition (speed-focus = 3.31, accuracy-focus = 3.33; t(59) = 0.612, p = 0.543; **Fig 3B**), a somewhat unusual finding since other research has consistently found an increase in confidence from the speed-focus to the accuracy-focus conditions [14,41]. Unlike the data in the current study but consistent with the previous literature, all models showed an increase in confidence from the speed-focus to accuracy-focus condition (Top2Diff: 3.16 vs. 3.58; ProbAvgRes: 3.22 vs. 3.50; ProbTop2Diff: 3.22 vs. 3.52; Entropy: 3.06 vs. 3.68; PE: 2.94 vs. 3.77; Softmax: 3.11 vs. 3.68; BCH: 3.29 vs. 3.51). Computing the average sum of squared errors across subjects showed that the ProbAvgRes model provided numerically the closest match to human confidence (Top2Diff = 0.155; ProbAvgRes = 0.121; ProbTop2Diff = 0.121; Entropy = 0.272; PE = 0.459; Softmax = 0.354; BCH = 0.142). These errors were significantly lower for the ProbAvgRes model compared to all other models except ProbTop2Diff and BCH (Top2Diff: t(59) = 5.170, p < .001; ProbTop2Diff: t(59) = 0.018, p = 0.986; Entropy: t(59) = 10.280, p < 0.001; PE: t(59) = 9.730, p < 0.001; Softmax: t(59) = 5.522, p < 0.001; BCH: t(59) = 1.521, p = 0.134).

Next, we assessed whether our human subjects and models generated the "folded-X pattern" [37,38], where, as the task gets easier, confidence for correct trials increases but confidence for error trials decreases. Indeed, human confidence followed this folded-X pattern, such that the easy condition exhibited higher confidence for correct trials (difficult = 3.39, easy = 3.66; t(59) = 9.727, p < 0.001; **Fig 3C**) but lower confidence for error trials (difficult = 2.74, easy = 2.68; t(59) = -2.095, p = 0.041). All strategies predicted that confidence for correct trials should be higher for the easy compared to the difficult condition (Top2Diff: difficult = 3.46, easy = 3.62; ProbAvgRes: difficult = 3.44, easy = 3.61; ProbTop2Diff: difficult = 3.44, easy = 3.61; Entropy: difficult = 3.43, easy = 3.56; PE: difficult = 3.40, easy = 3.47; Softmax: difficult = 3.46, easy = 3.60; BCH: difficult = 3.48, easy = 3.65). Similarly, all models correctly predicted that confidence for error trials should be lower for the easy compared to the difficult condition (Top2Diff: difficult = 2.96, easy = 2.82; ProbAvgRes: difficult = 2.91, easy = 2.87; ProbTop2Diff: difficult = 2.94, easy = 2.88; Entropy: difficult = 3.05, easy = 2.90; PE: difficult = 3.16, easy = 3.09; Softmax: difficult = 3.01, easy = 3.00; BCH: difficult = 3.95, easy = 3.92). Numerically, the ProbAvgRes model produced the most precise fits to the folded-X pattern observed in humans as measured by the sum of squared errors across subjects (Top2Diff = 0.245; ProbAvgRes = 0.238; ProbTop2Diff = 0.291; Entropy = 0.460; PE = 0.558; Softmax = 0.437; BCH = 0.308). However, these errors were not significantly different compared to the Top2Diff and ProbTop2Diff models (Top2Diff: t(59) = 0.433, p = 0.66; Top2Diff: t(59) = 1.822, p = 0.07) but were significantly lower compared to all other models (Entropy: t(59) = 3.437, p = 0.001; PE: t(59) = 8.165, p < 0.001; Softmax: t(59) = 3.482, p < 0.001; BCH: t(59) = 2.113, p = 0.038).

Overall, as may be expected from the AIC fits, the top three models which yielded the best AIC fits – Top2Diff, ProbAvgDiff, ProbTop2Diff – produced the most consistently accurate matches to the observed patterns of human confidence across all three qualitative patterns.

### Assessing model predictions of confidence for stimulus categories

Apart from capturing the behavioral patterns of different experimental manipulations, a robust model of confidence should also account for stimulus-driven variations. Specifically, certain categories of stimuli (such as the number 1) may be more prone to confusion with other digits (such as 7) and therefore, may be associated with lower confidence on average. Therefore, we assessed whether models could mimic the observed patterns of confidence across the eight stimulus categories. In addition to stimulus categories, we separated trials based on choice accuracy, since observed patterns of confidence depend on the accuracy of the primary choice. Confidence on error trials is generally lower compared to correct

trials and confidence for correct and error trials have opposing relationships with stimulus discriminability (**Fig 3C**). Mixing confidence across correct and error trials can thus obscure the underlying patterns.

First, we computed average confidence as a function of choice accuracy and stimulus category. Although confidence was generally well matched across the eight stimulus categories, there were small variations between categories. For correct trials, humans tended of give the highest confidence for the digit 8, and the lowest confidence for the digit 1. For error trials, the digit 8 was again associated with the highest confidence while 6 was associated with the lowest confidence. As expected, confidence was always higher for correct trials compared to error trials for humans as well as all models across all stimulus categories (**Fig 4**). Model comparisons showed that the Top2Diff model gave the closest fits to the overall pattern of confidence as assessed by the sum of squared errors (Top2Diff = 1.391; ProbAvgRes = 2.100; ProbTop2Diff = 2.035; Entropy = 2.427; PE = 3.402; Softmax = 2.088; BCH = 2.231). These errors were significantly lower for the Top2Diff model compared to all other models (ProbAvgRes: $t(59)$ = 7.970, $p < .001$; ProbTop2Diff: $t(59)$ = 6.124, $p < 0.001$; Entropy: $t(59)$ = 4.141, $p < 0.001$; PE: $t(59)$ = 9.922, $p < 0.001$; Softmax: $t(59)$ = 3.169, $p = 0.002$; BCH: $t(59)$ = 6.780, $p < 0.001$). In particular, for error trials, all other models produce substantial deviations from observed confidence for at least two stimulus categories, with the PE, Softmax, and Entropy models consistently overpredicting confidence across all stimulus categories.

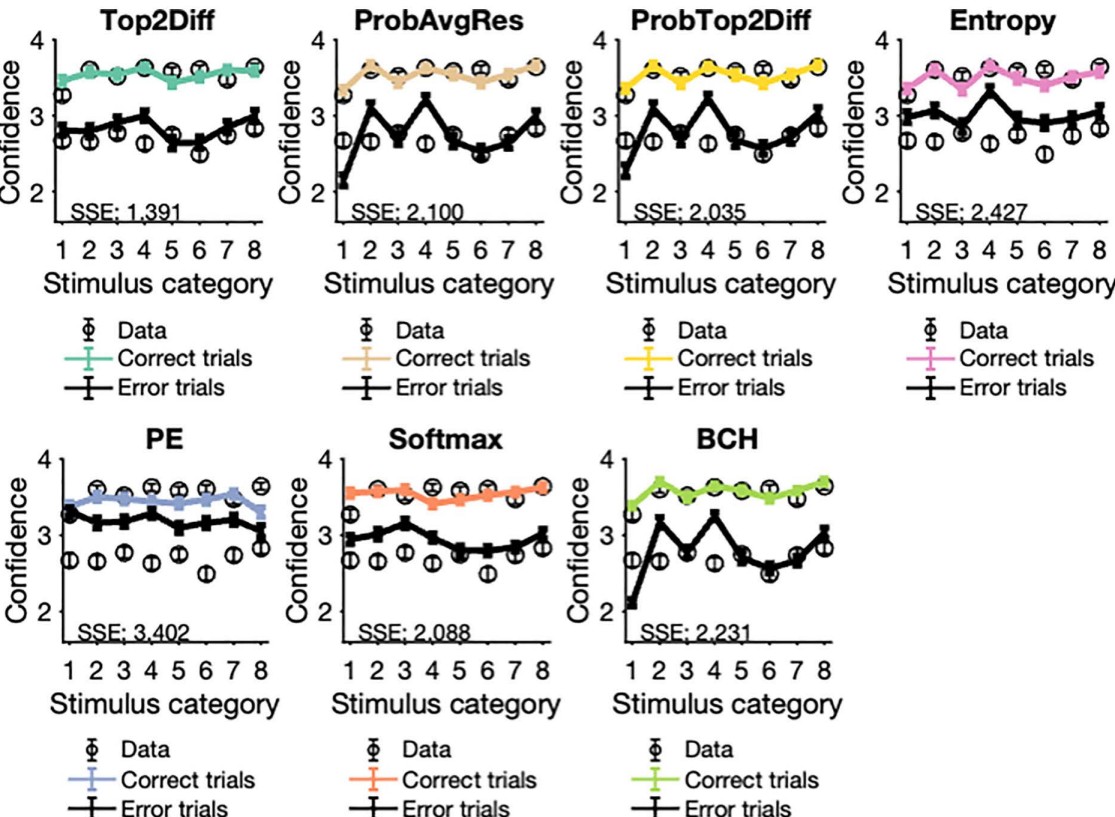

**Fig 4. Stimulus category-wise predictions of average confidence for correct and error trials.** Confidence was uniformly greater for correct trials compared to error trials across all eight stimulus categories. In addition, there were small variations in confidence across the eight categories, possibly due to some digits being more prone to confusion with others. All confidence strategies correctly predicted that confidence was higher for correct trials compared to error trials, but Top2Diff best captured the variations in confidence found between the stimulus categories. In particular, all models except Top2Diff over-estimated confidence for error trials for at least two categories, with the PE, Softmax, and Entropy models over-estimating confidence for all stimulus categories. Error bars depict SEM. SSE, sum of squared errors (smaller values indicate better fits).

Second, we analyzed how well models predicted the variations in confidence across stimulus categories for individual subjects. For each subject, we plotted averaged confidence within each stimulus category against the model's predicted confidence for that stimulus category, separately for correct and error trials. For correct trials (**Fig 5A**), all confidence strategies were able to predict individual confidence within the stimulus categories well – with predicted confidence falling within the observed range and showing a continuous increase within the range. None of the models showed a substantial tendency to over- or under-estimate confidence. However, for error trials (**Fig 6A**) the Entropy, PE, and Softmax models uniformly over-estimated confidence across stimulus categories as well as individuals, which may explain their poor fits, compared to the top-three models – Top2Diff, ProbAvgRes and ProbTop2Diff.

To quantify how well models could predict the variations in confidence between stimulus categories, we fit a linear mixed model which predicted human confidence for each stimulus category from the model's confidence for that category. Since the category-wise predictions of confidence were obtained separately for each individual subject, we controlled for the repeated measurements of these stimuli by including stimulus category as a random effect in the model. Using the mixed linear model's estimates of slope and intercept, we generated the model's predictions of category-wise confidence across individuals and correlated them with their corresponding observed quantities.

For correct trials, all strategies produced very strong correlations with human confidence (all r's > 0.871, all p's < 0.001; **Fig 5B**) across the eight stimulus categories. Comparing the Pearson's r-values showed that the Top2Diff model generated the numerically highest correlations, which were significantly higher than those produced by the Softmax and BCH models (Softmax: $z(479) = 3.789$, $p < 0.001$; BCH: $z(479) = 3.329$, $p < 0.001$). However, the Top2Diff model's correlations were not significantly higher than the ProbAvgRes, ProbTop2Diff, Entropy, and PE models (ProbAvgRes: $z(479) = .939$, $p = 0.348$; ProbTop2Diff: $z(479) = .196$, $p = .844$; Entropy: $z(479) = .196$, $p = .844$; PE: $z(479) = .576$, $p = .565$). In addition to these correlations, we also quantified the goodness-of-fit of the linear mixed model using AIC scores and found that the Top2Diff strategy generated the closest predictions of category-wise confidence for correct trials across individual subjects, with a difference of at least 10 AIC points with all other strategies (**Fig 5C**).

For error trials, the category-wise correlations between model and human confidence were slightly lower but nevertheless strong and significant for all confidence strategies (all r's > 0.818, all p's < 0.001; **Fig 6B**). Comparing the Pearson's r-values showed that the Top2Diff model once again generated the numerically highest correlations, which were significantly higher than those produced by the Entropy and Softmax models (Entropy: $z(479) = 2.100$, $p = 0.036$; Softmax: $z(479) = 2.627$, $p = 0.009$). However, the Top2Diff model's correlations were not significantly higher than those produced by the ProbAvgRes, ProbTop2Diff, PE, and BCH models (ProbAvgRes: $z(479) = 1.540$, $p = .124$; ProbTop2Diff: $z(479) = 1.644$, $p = .100$; PE: $z(479) = .1.110$, $p = .267$; BCH: $z(479) = .944$, $p = .345$). Comparison of AIC scores from the linear mixed model showed that the Top2Diff strategy also generated the best fits to category-wise confidence for error trials across individual subjects, with a substantial difference of at least 62 AIC points with all other strategies (**Fig 6C**). Together, these results suggest that the Top2Diff strategy best captures the patterns of variations in human confidence arising from dependencies in the stimulus categories.

## Discussion

We instantiated seven different confidence strategies in a recently developed neural network model of perceptual decisions, RTNet. We found that human confidence is best described by a strategy that computes confidence as the difference in evidence between the top-two choices (Top2Diff). The Top2Diff strategy not only produced the best quantitative fits to human data but also consistently gave one of the best fits to all qualitative signatures of human confidence and the best predictions of confidence by stimulus category and choice accuracy. These findings inform current debates on confidence. Specifically, they support the view that confidence is derived from raw evidence rather than probability computations and suggest that confidence uses a small and specific subset of evidence instead of all the available evidence. They also establish artificial neural networks as promising models for inferring the mechanisms underlying human perceptual and confidence judgments for complex images.

PLOS Computational Biology

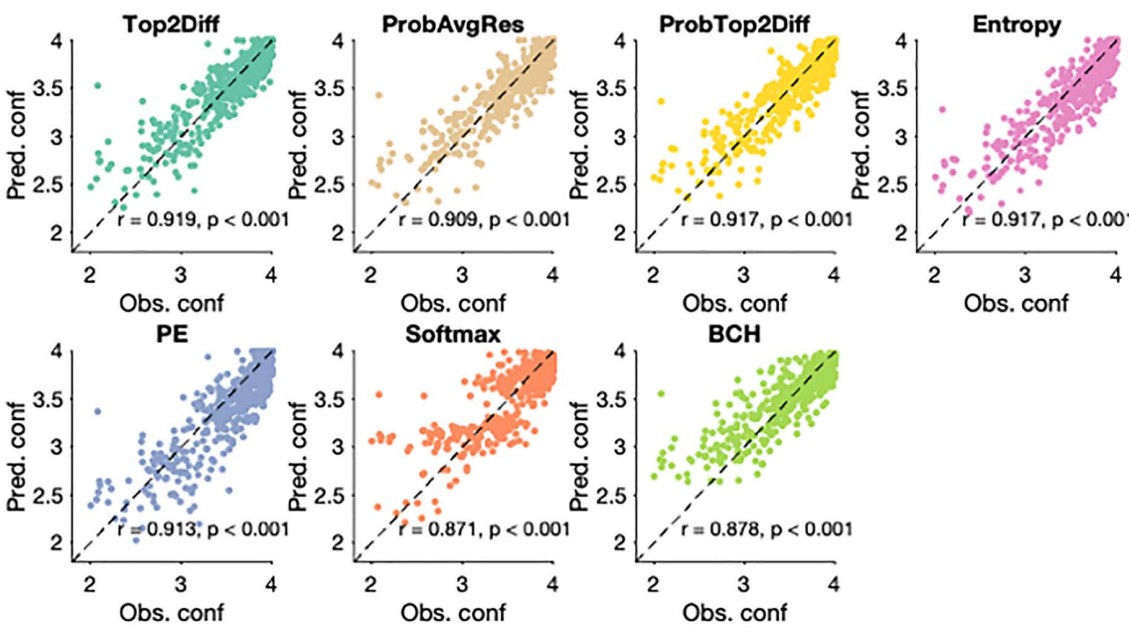

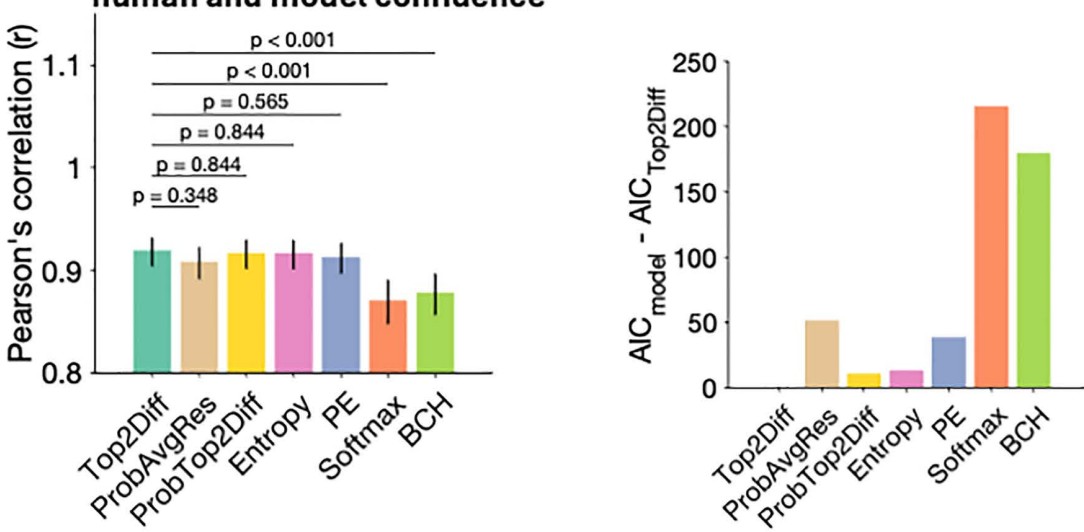

**Fig 5. Stimulus category-wise model predictions of individual confidence for correct choices.** (A) We computed average confidence for correct trials within each stimulus category separately for each individual subject and plotted these against the corresponding quantities predicted by each model. All models generated strong and significant correlations with observed confidence, suggesting that they were able to capture stimulus-related variations in confidence. (B) The Top2Diff strategy generated the numerically highest correlations with human confidence. However, these correlations were not significantly higher than those generated by ProbAvgRes, ProbTop2Diff, Entropy, and PE models. Error bars show 95% confidence intervals. (C) We fit a linear-mixed model that quantified how closely each model captured observed category-wise variations in confidence, while controlling for the repeated measurement across individuals. AIC scores derived from the linear mixed models showed that the Top2Diff strategy generated the best fits to category-specific confidence for correct trials with an AIC difference of at least 10 points with all other strategies.

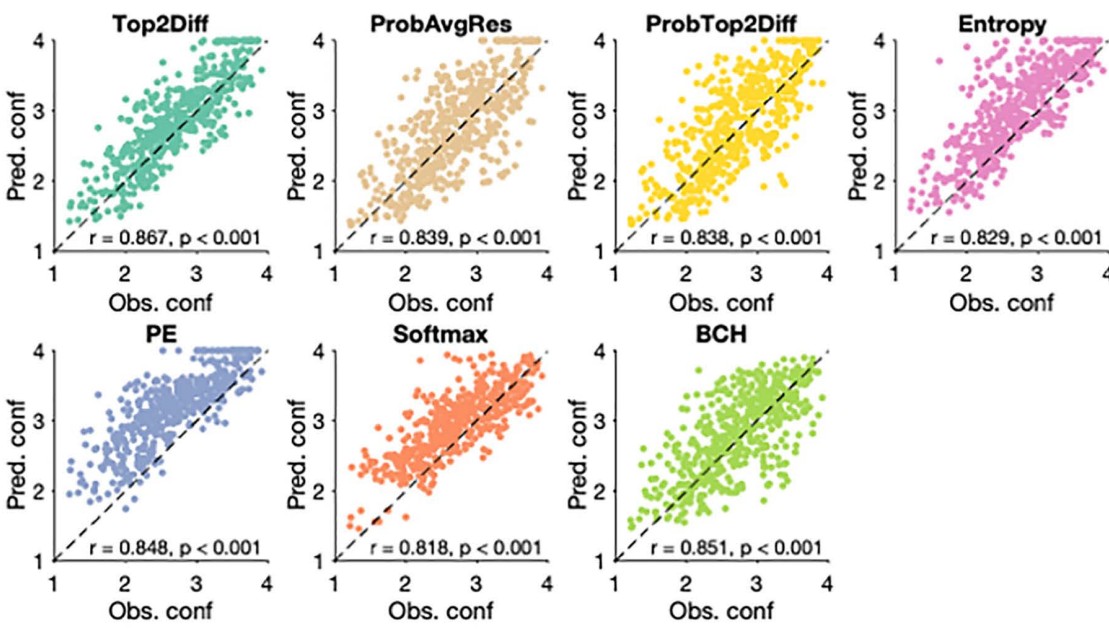

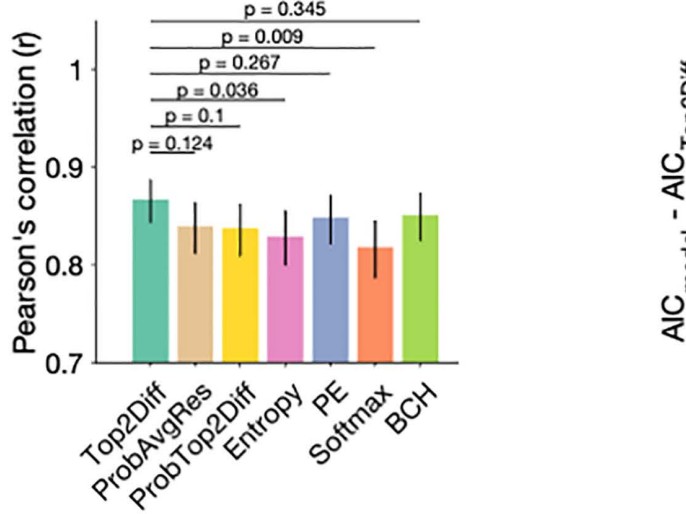

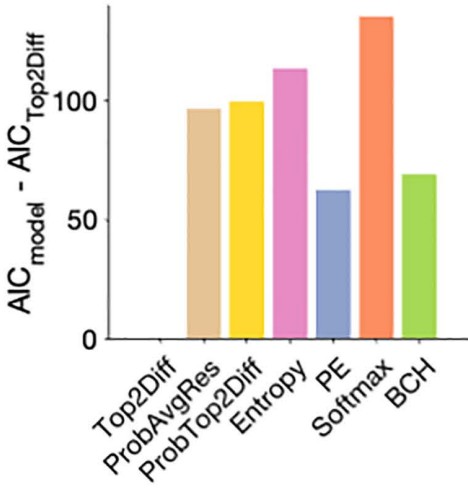

**Fig 6. Stimulus category-wise model predictions of individual confidence for error trials.** (A) We computed average confidence for error trials within each stimulus category separately for each individual subject and plotted these against the corresponding quantities predicted by each model. As with correct trials, models generated strong and significant correlations with observed confidence, suggesting that they are generally able to capture stimulus-related variations in confidence. (B) The Top2Diff strategy generated the numerically highest correlations with human confidence, but these correlations were not significantly higher than those generated by ProbAvgRes, ProbTop2Diff, PE and BCH models. Error bars show 95% confidence intervals. (C) We fit a linear-mixed model that quantified how closely each model captured observed category-wise variations in confidence, while controlling for the repeated measurement across individuals. AIC scores from these regression models showed that the Top2Diff strategy generated the substantially better fits to category-specific confidence for error trials with an AIC difference of at least 62 points with all other strategies.

Why does human confidence resemble the Top2Diff strategy despite the suboptimality of its underlying computation? One possibility is that the Top2Diff strategy may be preferred over more optimal strategies that use all available information since these optimal strategies can require very complex computations [42]. Particularly, in situations where there are many alternatives and the possible choices are not clearly defined, posterior probability computations become intractable as they require the system to compute probabilities associated with all possible choices [23]. On the other hand, the PE heuristic minimizes computational effort but neglects too much available information, thus producing confidence that may not be sufficiently informative about one's perceptual accuracy. Therefore, the Top2Diff strategy may have emerged as a viable solution to the trade-off between maximizing the information content of confidence ratings and minimizing computational costs.

It is important to note that the Top2Diff strategy does not imply that information about the other alternatives is completely lost since the evidence generated for all the choice alternatives is driven by the same stimulus and likely to be correlated. In that sense, knowing the evidence for even just two alternatives means that one has partial knowledge about the likely evidence for the other alternatives. The Top2Diff model simply assumes that the computation itself does not explicitly consider the evidence for those other alternatives.

The Top2Diff strategy is consistent with the long history of cognitive models developed for two-choice tasks. In most such models – whether they are based on signal detection theory [7], the drift-diffusion model [14], or race models of evidence accumulation [43] – confidence reflects the difference in raw evidence of the two options. Our results are also in line with a recent preprint from Xue et al. [44] where the authors report that Top2Diff outperforms PE, Softmax, Entropy, BCH, and ProbTop2Diff in 3- or 5-choice tasks with colored dots stimuli. Our study extends these previous findings to more complex stimuli and tasks that involve many more choice options.

Our findings add to a growing body of evidence against the notion that the positive evidence heuristic underlies human confidence. In spite of its popularity [25,33,36], the PE mechanism has been challenged by numerous recent findings from cognitive and neural network modelling [9,13,45,46]. These studies have shown that the PE model consistently produces some of the worst fits to human data compared to other existing models and that the PE mechanism is, in fact, not necessary to account for behavioral effects that were previously assumed to be its signatures [25,33]. A popular argument in favor of the PE heuristic is that it is necessary to keep confidence computations tractable when there are indefinite choice options [25,36]. However, our results indicate that even in tasks with many alternatives, this heuristic fails to account for human confidence. The confidence computation instead appears to be better described by the Top2Diff computation – a strategy of similar complexity that carries more information.

Our results support the view that human confidence judgments are unlikely to reflect complex probability computations [9,13,18–20,22,39]. Previous studies on confidence in multi-alternative perceptual tasks only compared probability-based models [22,24]. However, recent research shows that probability-based models rarely provide a good description of human confidence [47]. For example, several previous studies have strongly challenged the assumption that confidence follows the BCH strategy, which is the most prominent probability-based model [18,19,20,22]. It is therefore critical for future studies of confidence to always consider models where confidence is based directly on the sensory evidence and not solely focus on probability-based confidence models.

The standard practice in neural network modeling is to derive confidence using the Softmax (and less often the Entropy) computation. Our results demonstrate that this way of deriving confidence may not match how humans generate confidence. Nevertheless, these strategies likely produce more informative confidence ratings because they utilize all the information available from the primary decision. Therefore, Softmax and Entropy may be better suited to derive confidence from neural networks when the goal is to maximize insight into the network's performance rather than model human behavior.

One limitation of our study was the inability to fit the neural network models directly to choice and confidence data for individual subjects. Since neural networks have parameters in the order of millions, it becomes computationally intractable

to simultaneously tune all these parameters and fit the observed human data. Therefore, we used a general-purpose training method to teach the models to perform the task and then optimized two critical parameters that controlled the models' performance levels. Even then, we fit these two parameters to the average human data (instead of each subject separately) because RTNet requires significant computational resources and processing time to generate responses even on a single trial. Instead, the only parameters we could fit on a subject-by-subject basis using traditional exhaustive search that is standard in cognitive models [48,49] were the three confidence criteria. Since all the confidence strategies were fit for the same model of choice, these limitations are not expected to significantly impact the results we report here. Future work should focus on developing methods that would allow traditional maximum likelihood estimation to be applied so that neural network models can fit individual subjects.

A limitation of using RTNet as a model of human perceptual decisions is that RTNet underestimates the stochasticity of human decisions. When presented with the same stimulus multiple times, humans tend to make more inconsistent choices than RTNet. However, in spite of this limitation, RTNet's RTs and confidence correlated significantly with those of individual subjects [28], suggesting that RTNet can still function as sufficiently a good model of human choice and RT to allow meaningful comparison of confidence strategies.

A final limitation of our study is that the complexity of the task may have made it difficult to realize the speed-accuracy trade-off manipulation. Our task structure was relatively complex – featuring eight choice options and a 2 x 2 factorial design with difficulty and SAT manipulations. Such a complex task structure may have reduced the effect of the SAT manipulation. Indeed, the SAT manipulation had a very small effect on accuracy (2–3%; significant for the easy stimuli but not for the difficult stimuli), making it less surprising that the effect on confidence was also very small (and not significant). Previous studies, on the other hand, have reported an increase in confidence from the speed-focus to the accuracy-focus conditions [14,41], suggesting that increases in confidence do indeed occur with SAT manipulations. All confidence strategies examined here led to an increase of confidence from the speed-focus to the accuracy-focus conditions. Given the considerations above, we believe that this divergence from the current data should not be seen as a limitation of the models but as a limitation of the current dataset.

In conclusion, we show that a novel neural network model of human decision making, RTNet, can be used to infer the mechanisms of confidence generation in humans for multi-choice tasks with relatively complex stimuli. The strategy which computes confidence as the difference in evidence between the top-two choices (Top2Diff) emerged as the clear winner over other strategies that base confidence on the whole distribution of evidence or neglect choice-incongruent evidence. Our results highlight neural network models as promising tools to test hypotheses about the mechanisms of decision making and metacognition under naturalistic settings.

## Methods

### Ethics statement

All subjects signed informed consent and were compensated monetarily for their participation. The protocol was approved by the Georgia Institute of Technology Institutional Review Board (protocol no. H15308).

### Stimuli and task

Sixty-four subjects (31 female; age, 18–32 years) with normal or corrected-to-normal vision participated in the experiment. This experiment has been previously reported in Rafiei et al. [28] and all the details can be found in the original publication. Below, we briefly describe the experimental design.

Subjects performed a digit discrimination task with eight choices and reported confidence. On each trial, a fixation cross was presented for 500–1000 ms followed the presentation of a stimulus for 300 ms. The stimulus was a digit between 1–8 and superimposed with noise. Subjects reported the perceived digit by pressing a key between 1–8 and subsequently

reported their confidence on a scale from 1-4 via another key press (where 1 corresponds to lowest confidence and 4 corresponds to highest confidence). The response screens stayed on until subjects made a response.

We manipulated task difficulty as well as speed-accuracy trade-off (SAT) instructions. Task difficulty was manipulated by corrupting the pixels with uniform noise. Easier stimuli contained lower pixel noise on average. SAT was manipulated by instructing subjects to make their responses as fast or as accurate as possible. Trials containing easy and difficult stimuli were interleaved whereas trials with different SAT instructions were blocked and presented alternately.

The stimuli were obtained from the publicly available MNIST dataset containing 60,000 training images and 10,000 testing images. The stimuli shown to subjects were taken only from the testing set to ensure that both subjects and networks were tested on novel images. 480 images were randomly selected and evenly distributed across the four experimental conditions. Subjects completed 4 blocks of 60 trials each (960 trials in total as each of the 480 images was presented twice). The experiment was designed in the MATLAB v.2020b environment using Psychtoolbox 3 [50]. The stimuli were presented on a 21.5-inch Dell P2217H monitor (1,920 × 1,080 pixel resolution, 60 Hz refresh rate). The subjects were seated 60 cm away from the screen and provided their responses using the keyboard.

## Behavioral analyses

We excluded subjects based on preregistered criteria (https://osf.io/kmraq). These criteria resulted in the exclusion of four subjects in total (two subjects for not following SAT instructions and two subjects for ceiling effects on confidence). For each individual subject, we computed average confidence as a function of task difficulty and SAT condition to assess how these factors affect confidence. We also assessed confidence ratings for the folded-X pattern, which is popularly regarded to be a signature of human confidence. The folded-X pattern is obtained by computing confidence as a function of task difficulty separately for correct and error trials.

## Network architecture and implementation

We first briefly describe the RTNet architectures and its implementation. We trained 60 instances of the network using different random initializations of the network's weights to allow for individual differences in learning. RTNet was implemented in Python (version 3.10.11).

The RTNet architecture consists of two modules. The first module is a Bayesian neural network (BNN) whose weights are learned as posterior probability distributions rather than point estimates [51]. Due to stochasticity in the BNN's weights, a unique feedforward network gets sampled at each time step and repeated processing of the same image generates variable activations in the network's final layer. The second module consists of an evidence accumulator that receives these noisy activations from the network's final layer and integrates the evidence towards a pre-defined threshold. Evidence is accumulated independently for each choice option and the model chooses the option for which the evidence first hits the threshold. Response time corresponds to the number of evidence samples that were required to reach the threshold. The network was implemented within the AlexNet architecture [52] consisting of five convolutional layers and three fully connected layers.

## Fitting RTNet to human choices

Our central goal was to compare model fits to human confidence across confidence strategies. However, due to computational constraints, it is not possible to simultaneously fit both choice and confidence data derived from RTNet via traditional maximum likelihood estimation (MLE) methods. Therefore, we followed a step-wise approach where we 1) trained RTNet on the general task performed by humans, 2) fit RTNet to human choices by optimizing two additional parameters (noise in the stimuli and decision threshold) that gave us the closest match to the average human performance, and 3) generated confidence using each of the seven strategies and fit three confidence criteria separately for each individual

human subject in order to create subject-specific fits for the confidence ratings. In Step 1, we trained RTNet on the MNIST dataset to classify handwritten digits. RTNet was trained to achieve an accuracy of >97% on the training set. The details of the training procedure can be found in the original publication. Training RTNet and tuning the choice parameters (Steps 1 and 2) were previously done by Rafiei et al. [28]. In the current study, we simulated these pre-trained and tuned networks to generate confidence using seven different strategies (Step 3). Importantly, in Step 2, we fit two choice parameters which were the same for all human subjects, but in Step 3, we fit individual confidence criteria that differed from subject to subject. Below, we describe the procedures for Steps 2 and 3 in more detail.

### Fitting the pre-trained models to human choices (Step 2 of fitting)

To fit the models to human choices, we matched RTNet's accuracy levels to those observed in humans by tuning two additional parameters. We separately matched the network's performance in each experimental condition to the average accuracy observed for that condition. To match accuracies across the difficulty levels, we adjusted the noise in the images. Images with higher levels of noise lead to lower accuracy. To mimic the effect of the speed-accuracy trade-off manipulation on accuracy, we adjusted the RTNet's thresholds. A higher threshold leads to longer processing times and higher accuracies.

We estimated the parameters using a coarse search followed by a fine-grained search and chose the parameters that gave us the closest match to average human accuracy for each condition. The best match to human accuracy was achieved for noise levels of 2.1 for easy images and 4.1 for difficult images, and threshold values of 3 for the speed condition and 6 for the accuracy condition.

### Fitting the model responses to human confidence (Step 3 of fitting)

**Implementing different confidence strategies.** Confidence is computed from the evidence at the final time step at which the network makes a decision. We implemented seven confidence strategies: PE, Top2Diff, Softmax, BCH, ProbTop2Diff, ProbAvgRes, and Entropy. We first obtained the accumulated evidence, that is, the logits or raw activations for each of the eight choice options, $\mathbf{z} = [z_1, z_2, \ldots z_8]$. The positive evidence hypothesis posits that confidence is based on the evidence in favor of the chosen option. Therefore, under the PE model, confidence was computed as the activation associated with the chosen response, such that $conf_{PE} = \max(\mathbf{z})$. According to the Top2Diff model, confidence was computed as the difference in activation between the chosen response and the response that generated the second highest evidence, such that $conf_{Top2Diff} = \max(\mathbf{z}) - max2(\mathbf{z})$ where $max2(\mathbf{z})$ is the second highest value in the vector $\mathbf{z}$. Since the activations are unbounded, they can take arbitrarily large values which can be problematic when fitting models to the data. Therefore, for both the PE and Top2Diff models, the raw activation scores across trials were normalized by their standard deviations to restrict their ranges. For the Softmax model, we first applied the softmax transformation to the raw activations, such that $s_i = \frac{e^{z_i}}{\sum_{j=1}^{8} e^{z_j}}$, where $s_i$ is the softmax value associated with the $i^{th}$ choice. The softmax function transforms the activations into probability scores where $s_i \in [0, 1]$ and $\sum_{i=1}^{8} s_i = 1$. Softmax confidence was then obtained as $conf_{Softmax} = \max(s_i)$.

The BCH, ProbTop2Diff, ProbAvgRes, and Entropy models posit that confidence is derived from the posterior probability associated with each alternative rather than the raw evidence. To obtain these posterior probabilities, one has to be able to compute the likelihood of the observed raw activations on a given trial for each category, $C_i$. To be able to compute these likelihoods, we first generated a large set of internal activations using 9520 held-out MNIST validation images (excluding the experimental stimuli). For each category $i$, we modeled the likelihood function for each stimulus category as a multivariate normal distribution: $\mathcal{N}(\mu_i, \sum_i)$, where $\mu_i$ and $\sum_i$ are the mean vector and covariance matrix for class $i$. Having derived these likelihood functions, we could then compute the posterior probability associated with each alternative given a new set of raw evidence $\mathbf{z}$ by applying Bayes' theorem (and taking into account that all stimulus categories have equal prior probabilities):

$$p_i = \frac{\mathcal{N}(\boldsymbol{z} \mid \mu_i, \sum_i)}{\sum_{i=1}^{8} \mathcal{N}(\boldsymbol{z} \mid \mu_i, \sum_i)}$$

This procedure converts raw activations $\boldsymbol{z}$ into posterior probabilities $\boldsymbol{p}$, where $p_i \in [0, 1]$ and $\sum_{i=1}^{8} p_i = 1$, which were then used to compute confidence for the four probability-based models. Specifically, the BCH model posits that confidence reflects the probability that the chosen option is correct, given the evidence. Thus, confidence for the BCH model is computed as the posterior probability associated with the chosen response: $conf_{BCH} = p_{chosen}$. The ProbTop2Diff model is the probability version of the Top2Diff computation, such that confidence is computed as the difference in posterior probability between the chosen response and second most probable choice [22]: $conf_{ProbTop2Diff} = p_{chosen} - p_{second-choice}$. For the ProbAvgRes model, confidence is proposed to be the difference between the posterior probability of the chosen response and the mean of the remaining posterior probabilities [24] such that $conf_{ProbAvgRes} = p_{chosen} - \frac{1}{7} \sum_{i \neq chosen} p_i$. Finally, for the Entropy model, the entropy, $H$, is computed based on the posterior probability $H = -\sum_{i=1}^{8} p_i * \log(p_i)$. Since confidence should increase with decreasing uncertainty or entropy, we defined confidence as the negative of entropy, such that $conf_{Entropy} = -H$.

In all these computations, it is assumed that confidence arises from the final processing layer of RTNet. It is in principle possible to derive confidence using information in all layers of RTNet rather than the information in the last layer only. However, implementing this possibility would require one to interpret evidence from all the nodes of a neural network, which is a highly complex computation that may be practically unfeasible. For instance, there are thousands of nodes even within a single layer of the network and it is unclear how these nodes relate to the evidence for a given option. Further, these networks are structured in a way that the final layer is the one that is maximally informative about the evidence for each option and considering the previous layers may only add noise to the computation. Therefore, for these neural network models, we assume that confidence is derived entirely from the evidence in the final layer of the network.

**Performing model fitting.** Human confidence responses were obtained on a four-point scale. However, the models' confidence was derived from their raw and continuous activations, making it difficult to directly compare the models' confidence to humans. Therefore, we ran a model fitting procedure to fit the models' confidence responses separately for each individual human subject. We defined a set of three confidence criteria for each subject that would transform the continuous confidence responses into subject-specific 4-point confidence, thus allowing direct comparisons with humans.

Model fitting was based a maximum likelihood estimation (MLE) procedure that aimed to find a set of parameters that maximized the log-likelihood of the model associated with the full probability distribution of responses. The probability distribution of responses was computed as an 8 x 8 x 4 response matrix representing the eight stimulus classes and 8 x 4 responses (eight choices with four confidence ratings for each choice). For each model, we pooled the responses across the 60 model instances and binned them according to their associated stimulus, choice, and confidence to obtain the response probability matrix. Log-likelihood was computed as: $\sum_{i,j,k} \log(p_{ijk}) \times n_{ijk}$, where $p_{ijk}$ refers to the response probability (computed from the model's simulated responses) and $n_{ijk}$ refers to the number of trials associated with stimulus class $i = \{1, 2, \ldots, 8\}$, choice $j = \{1, 2, \ldots, 8\}$ and, confidence response $k = \{1, 2, 3, 4\}$. The parameter search was conducted using the Bayesian Adaptive Direct Search (BADS) toolbox [48]. Quantitative model comparisons were made using a goodness-of-fit measure called the Akaike Information Criterion (AIC) which is computed as $AIC = -2logL + 2k$, where $logL$ refers to the log-likelihood associated with the maximum-likelihood estimates of the parameters and $k$ refers to the number of model parameters. For all models, $k$ was fixed as three (corresponding to the three confidence criteria) because we were only fitting the model's responses to confidence.

**Generating the model's confidence responses for individual subjects.** After obtaining the MLE of the confidence criteria for each individual subject, we used these parameters to simulate subject-specific confidence ratings. To generate these ratings, we first derived raw confidence for each trial by applying the confidence generation strategies defined above. To generate subject-specific confidence, we applied that subject's estimated confidence criteria to the raw

confidence values after aggregating the networks' output across all 60 instances. The confidence criteria were defined as $c_{conf} = [c_0, c_1, c_2, c_3, c_4]$, where $c_0 = -\infty$ and $c_4 = \infty$, and were applied to the raw confidence values ($r_{conf}$) such that $r_{conf}$ falling within the interval $[c_{i-1}, c_i)$ resulted in a confidence rating of $i$ where $i \in \{1, 2, 3, 4\}$.

**Qualitatively assessing models' confidence responses and predictions**

We analyzed whether the models' confidence ratings reproduced the qualitative patterns of confidence observed in humans. Specifically, we simulated each subject's confidence separately using their individual parameter estimates and analyzed whether these simulated confidence ratings reproduced the observed effects of the difficulty manipulation and the SAT manipulation. We also examined the "folded-X pattern" of confidence where, as the task gets easier, confidence for correct trials increases but confidence for error trials decreases [53–55]. Mean confidence for each condition was computed separately for each simulated subject and then averaged across subjects.

For completeness, we performed similar model comparisons using three other CNN architectures – MSDNet (multi-scale dense network; [31]), BLNet (a recurrent convolutional network; [29]) and CNet (a parallel cascaded network; [30] – that were previously trained on MNIST and fit to human choice data [28]. These networks provide relatively poor fits to the human choice and RT data [28], making any confidence model comparisons built on them less informative. Nevertheless, we report confidence model comparisons within these architectures in S1 Text and S1 Fig.

## Supporting information

**S1 Text.**
(DOCX)

**S1 Fig. Comparing model fits across all CNN architectures for four confidence strategies.** (A) Quantitative comparisons using AIC scores. Positive AIC differences indicate support for the RTNet-Top2Diff model. We first assessed the models' fits to the data by computing average AIC values across the 60 subjects for the 16 models (4 architectures x 4 confidence strategies). We found that the confidence strategies built on top of RTNet generated substantially lower AIC values compared to the remaining 12 models (3 architectures x 4 confidence strategies). The Top2Diff strategy instantiated within RTNet remained the best model, with other strategies instantiated within RTNet falling between 45–170 AIC points behind it. In contrast, the four confidence strategies instantiated within MSDNet were 430–778 points worse than the RTNet-Top2Diff model. BLNet and CNet produced even worse fits with strategies instantiated in BLNet falling behind the RTNet-Top2Diff model by 2860–3087 AIC points and strategies within CNet falling behind RTNet-Top2 Diff by 3245–3528 AIC points. All AIC comparisons were significant as assessed by computing bootstrapped 95% confidence intervals. These results corroborate our previous findings that RTNet provides the best fits to human data [28]. (B) The poor fits to data by other CNN architectures suggest that these architectures are unable to simultaneously fit the observed patterns of choices and confidence. However, since fitting models to choices and confidence was done in separate steps, it is important to rule out the possibility that these poor fits are due to a failure of models to fit confidence alone. Therefore, we correlated each subject's average confidence to the model's average confidence for that subject. We found that all models and strategies produced very high correlations (median r = 0.934, all r's > 0.75, all p's < 0.001), confirming that the fitting procedure was successful. Instead, the poor fits for MSDNet, BLNet, and CNet were likely a result of the failure of these architectures to fit the pattern of human choices underlying confidence, in spite of being able to fit the overall confidence for each individual subject well.
(TIFF)

## Author contributions

**Conceptualization:** Medha Shekhar, Herrick Fung, Krish Saxena, Farshad Rafiei, Dobromir Rahnev.

**Data curation:** Farshad Rafiei.

**Formal analysis:** Medha Shekhar, Herrick Fung, Krish Saxena.

**Funding acquisition:** Dobromir Rahnev.

**Investigation:** Medha Shekhar, Herrick Fung, Krish Saxena, Dobromir Rahnev.

**Methodology:** Medha Shekhar, Herrick Fung, Krish Saxena, Farshad Rafiei.

**Project administration:** Dobromir Rahnev.

**Resources:** Farshad Rafiei.

**Software:** Medha Shekhar, Farshad Rafiei.

**Validation:** Dobromir Rahnev.

**Visualization:** Medha Shekhar.

**Writing – original draft:** Medha Shekhar.

**Writing – review & editing:** Herrick Fung, Krish Saxena, Farshad Rafiei, Dobromir Rahnev.

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
