## [Decision Letter · Decision Letter 0]

17 Apr 2025

PCOMPBIOL-D-25-00064

Using artificial neural networks to reveal the human confidence computation

PLOS Computational Biology

Dear Dr. Shekhar,

Thank you for submitting your manuscript to PLOS Computational Biology. After careful consideration, we feel that it has merit but does not fully meet PLOS Computational Biology's publication criteria as it currently stands. Therefore, we invite you to submit a revised version of the manuscript that addresses the points raised during the review process. 

You will see both reviewers found the work interesting, but pointed out some issues with the manuscript. In particular, I will draw your attention to the comments of reviewer 2 about the predictions of the models tested and the misalignment with human data. In particular, what does it mean for conclusions made that all models misaligned with humans? 

Please submit your revised manuscript within 60 days Jun 17 2025 11:59PM. If you will need more time than this to complete your revisions, please reply to this message or contact the journal office at ploscompbiol@plos.org. Please include the following items when submitting your revised manuscript:

We look forward to receiving your revised manuscript.

Kind regards,

Alex Leonidas Doumas

Academic Editor

PLOS Computational Biology

Lyle Graham

Section Editor

PLOS Computational Biology

**Journal Requirements:**

1) Please provide an Author Summary. This should appear in your manuscript between the Abstract (if applicable) and the Introduction, and should be 150-200 words long. The aim should be to make your findings accessible to a wide audience that includes both scientists and non-scientists. Sample summaries can be found on our website under Submission Guidelines:

4) Please amend your detailed Financial Disclosure statement. This is published with the article. It must therefore be completed in full sentences and contain the exact wording you wish to be published.

2) State what role the funders took in the study. If the funders had no role in your study, please state: "The funders had no role in study design, data collection and analysis, decision to publish, or preparation of the manuscript.".

**Reviewers' comments:**

Reviewer's Responses to Questions

Reviewer #1: This paper makes a clear case for a simple point: comparing top 2 options gives the best prediction of confidence ratings. The authors use multiple different analyses to support this point, including individual and group comparisons and exploring other network architectures. The claim is also consistent with cited past work.

What was lacking in the paper was an analysis of the choices made by the models. Specifically, are these confidence ratings taken only from images where the subject and the model make the same choice? It is hard to think about how to compare the model's confidence in an incorrect choice to a subject's confidence in their correct choice or vice versa. Furthermore, do the different architectures tested have better or worse correspondence to human choice on an image-by-image basis? This could be relevant for understanding their confidence rating differences (this is hinted at in the results but I don't think it was reported on directly).

Do the authors think that performing step 2 training on individual data would impact the results in any way?

The authors find no confidence difference between speed-focused and accuracy-focused trials. Can they remark on if this is consistent with prior studies?

"However, while this type of computation may not match well what humans do, it must be

noted that it produces more informative confidence ratings." What does this mean and what is it based on?

Do the authors have anything to note about what aspects of the data that the models still don't match, in order to inspire future research?

Clarification:

I think it would be useful for readability to have some more description of the model architectures and training the main Results.

Please add clarification to the methods about what timepoint/layer the logits/raw activations are taken from for confidence calculations. (I'm assuming it is the timepoint where the model reached the threshold or the layer they did, depending on model architecture).

typo: "Second, we evaluated the effect of the speed-accuracy trade-off manipulation on

confidenceClick or tap here to enter text.."

Reviewer #2: The current report examines the best way to compute confidence in deep neural networks, in order to mimic human confidence judgments. From a technical perspective this excellent work, and likewise the manuscript is very well written. From a conceptual standpoint, however, the novel insights are less clear to me. Most researchers interested in human confidence would not be surprised that softmax is not a good rule to capture human confidence (I’m not aware of any theory making such claims) and likewise most researchers working with DNNs would not be interested in whether they model human confidence (but instead would want the most informative measure).

1. For metacognition researchers the conclusion that human confidence is not well explained by softmax or entropy is not very surprising. First, it is the same conclusion reached by Li & Ma, admittedly using a more complex model that works with actual images. Also, I am not aware of any theory of confidence arguing that humans compute confidence using softmaxt or entropy? Second, a recent paper by Comay et al. 2023 (Cognition) showed tat confidence depends on a dud-alternative and proposed a average-residual account for their findings. It would be important to also test this strategy, and perhaps see whether the top2 rule can account for the dud effect (given that this seems to directly contradict predictions from this rule).

2. The authors model confidence based on the (accumulated) activation in the output layer, and call this evidence. I have some concerns about this. First, the top2 rule seems to imply that humans do not consider the evidence of other options, but this conclusion is only valid if the nodes in the output layer are independent. If that is not the case, and especially for MNIST it is not hard to conceive that evidence for 8 and 9 will not be independent from each other, the top2 rule could be an indirect way to consider all evidence. Second, I wonder whether this is the most sensible way to think about evidence, for example compared to a metacognitive module having access to the full processing stream of evidence. Why is confidence only calculated based on the final output node?

3. It is a bit worrisome that one of the three key empirical findings in the human data are not captured by (any of) the model(s). Humans do not give higher confidence in accuracy vs speed conditions, yet all models predict this. This is a rather glaring misfit that the authors seem to dismiss rather easily. Does this not point to a clear issue with the model(s)?

4. One of the first papers examining human confidence in DNNs, by Webb and colleagues (2023) is only briefly mentioned in the discussion as a paper implementing softmax confidence. My reading from this paper, however, is that confidence is implemented as a separate head which learns to compute confidence via reinforcement learning. As such, we don’t really know whether or not it learned to use softmax, or whether perhaps it learned to use the top2 diff rule. Likewise in the discussion Rafiei et al. (2024) is dismissed as using the softmax rule, but in that paper it reads “The confidence of the model was obtained by taking the difference in evidence scores between the chosen response and the second-best choice.” Is this not the top2 rule?

5. As a minor point: it would be interesting to get some insights about how all of this work relates to related human confidence literature. For example, is the top 2 rule consistent with confidence as a distance-to-criterion signal?

**Have the authors made all data and (if applicable) computational code underlying the findings in their manuscript fully available?**

Reviewer #1: None

Reviewer #2: Yes

PLOS authors have the option to publish the peer review history of their article (what does this mean? ). If published, this will include your full peer review and any attached files.

**Do you want your identity to be public for this peer review?** For information about this choice, including consent withdrawal, please see our Privacy Policy .

Reviewer #1: No

Reviewer #2: No

**Figure resubmission:**
---

## [Decision Letter · Decision Letter 1]

9 Dec 2025

Dear Dr. Shekhar,

We are pleased to inform you that your manuscript 'Using artificial neural networks to reveal the human confidence computation' has been provisionally accepted for publication in PLOS Computational Biology. Let me also thank you on behalf of myself and the reviewers for your very careful revision. You did an exemplary job clearly and thoroughly addressing the raised points in the cover letter, and reflecting those changes in the manuscript.  

Best regards,

Alex Leonidas Doumas

Academic Editor

PLOS Computational Biology

Lyle Graham

Section Editor

PLOS Computational Biology

Reviewer's Responses to Questions

**Comments to the Authors:**

Reviewer #1: I appreciate the substantial additional analyses the authors have performed. It has helped to address my questions.

Reviewer #2: This is an excellent revision. The inclusion of several alternative models makes the paper much more interesting and compelling. It remains a bit unfortunate that the SAT manipulation provides such discrepant results between model and data (while I agree with the reply that the data are somewhat unusual, that of course does not explain the discrepancy) but this has been somewhat addressed. Congrats on an excellent revision.

**Have the authors made all data and (if applicable) computational code underlying the findings in their manuscript fully available?**

Reviewer #1: None

Reviewer #2: Yes

PLOS authors have the option to publish the peer review history of their article (what does this mean? ). If published, this will include your full peer review and any attached files.

**Do you want your identity to be public for this peer review?** For information about this choice, including consent withdrawal, please see our Privacy Policy .

Reviewer #1: No

Reviewer #2: No

---

## [Editor Report · Acceptance letter]

PCOMPBIOL-D-25-00064R1

Using artificial neural networks to reveal the human confidence computation

Dear Dr Shekhar,

I am pleased to inform you that your manuscript has been formally accepted for publication in PLOS Computational Biology. Your manuscript is now with our production department and you will be notified of the publication date in due course.

With kind regards,

Anita Estes
